# RH20T: A Comprehensive Robotic Dataset for Learning Diverse Skills in One-Shot

**Abstract:** A key challenge for robotic manipulation in open domains is how to acquire diverse and generalizable skills for robots. Recent progress in one-shot imitation learning and robotic foundation models have shown promise in transferring trained policies to new tasks based on demonstrations. This feature is attractive for enabling robots to acquire new skills and improve their manipulative ability. However, due to limitations in the training dataset, the current focus of the community has mainly been on simple cases, such as push or pick-place tasks, relying solely on visual guidance. In reality, there are many complex skills, some of which may even require both visual and tactile perception to solve. This paper aims to unlock the potential for an agent to generalize to hundreds of real-world skills with multimodal perception. To achieve this, we have collected a dataset comprising over 110,000 *contact-rich* robot manipulation sequences across diverse skills, contexts, robots, and camera viewpoints, all collected *in the real world*. Each sequence in the dataset includes visual, force, audio, and action information. Moreover, we also provide a corresponding human demonstration video and a language description for each robot sequence. We have invested significant efforts in calibrating all the sensors and ensuring a high-quality dataset.

**Keywords:** Dataset, Robotic manipulation, Skill learning

## 1 Introduction

Robotic manipulation requires the robot to control its actuator and change the environment following a task specification. Enabling robots to learn new skills with minimal effort is one of the ultimate goals of the robot learning community. Recent research in one-shot imitation learning [1, 2] and emerging foundation models [3, 4] draw an exciting picture of transferring trained policies to a new task given a demonstration. This paper shares the same aspiration.

While the future is promising, most research in robotics only demonstrates the effectiveness of their algorithms on simple cases, such as pushing, picking, and placing objects in the real world. Two main factors hinder the exploration of more complex tasks in this direction. Firstly, there is a lack of large and diverse robotic manipulation datasets in this field [3], despite the community's long-standing eagerness for such datasets. The fundamental problem stems from the huge barriers associated with data acquisition. These challenges include the arduous task of configuring diverse robot platforms, creating varied environments, and gathering manipulation trajectories, which require significant effort and resources. Secondly, most methods focus solely on visual guidance control, yet it has been observed in physiology that humans with impaired digital sensibility struggle to accomplish many daily manipulations with visual guidance alone [5]. This indicates that more sensory information should be considered in order to learn various manipulations in open environments.

To address these problems, we revisit the data collection process for robotic manipulation. In most imitation learning literature, expert robot trajectories are manually collected using simplified user

Submitted to the 7th Conference on Robot Learning (CoRL 2023). Do not distribute.

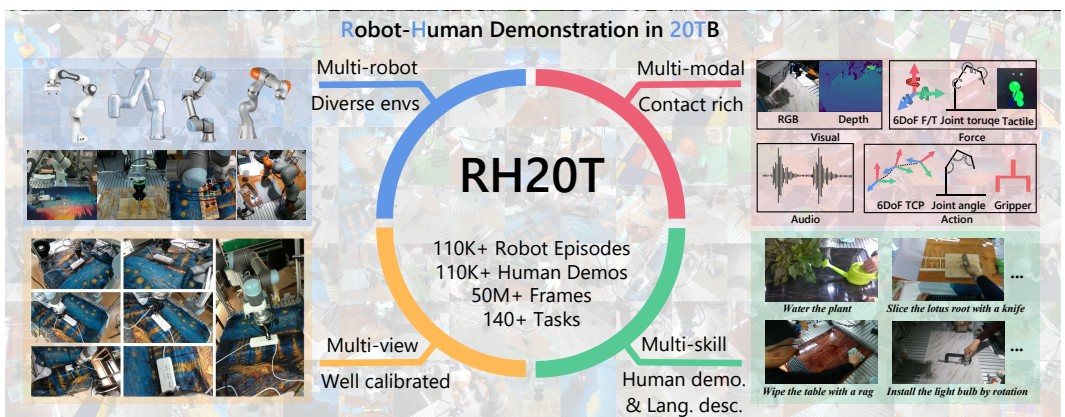

Figure 1: Overview of our RH20T dataset. We adopt multiple robots and setup diverse environments for the data collection. The robot manipulation episodes include multi-modal visual, force, audio and action data. For each episode, we collect the manipulation process with well calibrated multi-view cameras. Our dataset contains diverse robotic manipulation skills and each episode has a corresponding human demonstration and language description. In total, we provide over 110K robot episodes and 110K corresponding human demonstration. The dataset contains over 50 million frames and over 140 tasks.

interfaces like 3D mice, keyboards, or VR remotes. However, these control methods are inefficient and pose safety risks when the robot engages in rich-contact interactions with the environment. The main reasons are the unintuitive nature of controlling with a 3D mouse or keyboard, and the inaccuracies resulting from motion drifting when using a VR remote. Additionally, tele-operation without force feedback degrades manipulation efficiency for humans. In this paper, we equipped the robot with a force-torque sensor and employed a haptic device with force rendering for precise and efficient data collection. With the goal that the dataset should be representative, generalized, diverse and close to reality, we collect around 150 skills with complicated actions other than simple pick-place. These skills were either selected from RLBench [6] and MetaWorld [7], or proposed by ourselves. Many skills require the robot to engage in contact-rich interactions with the environment, such as cutting, plugging, slicing, pouring, folding, rotating, etc. We have used multiple different robot arms commonly found in labs worldwide to collect our dataset. The diversity in robot configurations can also aid algorithms in generalizing to other robots.

So far, we have collected around 110,000 sequences of robotic manipulation and 110,000 corresponding human demonstration videos for the same skills. This amounts to over 40 million frames of images for the robotic manipulation sequences and over 10 million frames for the human demonstrations. Each robot sequence contains abundant visual, tactile, audio, and proprioception information from multiple sensors. The dataset is carefully organized, and *we believe that a dataset with such diversity and scale is crucial for the future emergence of foundation models in general skill learning,* as promising progress has been witnessed in the NLP and CV communities [8, 9, 10].

## 2 Related Works

We briefly review related works in robotic manipulation datasets, zero/one-shot imitation learning, and vision-force learning methods.

**Dataset** Our community has been striving to create a large-scale and representative dataset for a significant period of time. Previous research in one-shot imitation learning has either collected robot manipulation data in the real world [2] or in simulation [11]. However, their datasets are usually small and the tasks are simple. Some attempts have been made to create large-scale real robot manipulation datasets [12, 13, 14, 15, 16, 17]. For example, RoboTurk [16] developed a crowd-sourcing platform and collected data on three tasks using mobile phone-based tele-operation.

MIME [17] collected 20 types of manipulations using Baxter with kinesthetic teaching, but they were limited to a single robot and simple environments. RoboNet [12] gathered a significant amount of robot trajectories with various robots, grippers, and environments. However, it mainly consists of random walking episodes due to the challenges of performing meaningful skills. BC-Z [14] presents a manipulation collection of 100 "tasks", but as pointed out in [11], they are combinations of 9 verbs and 6-15 objects. Similarly, RT-1 [4] and RoboSet [18] also collect large-scale manipulation datasets but focus on a limited set of skills. Concurrently to our work, BridgeData V2 [19] collects a dataset with 13 skills across 24 environments. In this paper, we present a larger dataset with a wider range of skills and environments, with more comprehensive information. More importantly, all previous datasets put less emphasize on contact-rich manipulation. Our dataset focus more in this case and include the crucial force modality during manipulation.

**Zero/One-shot imitation learning**   The objective of training policies that can transfer to new tasks based on robot/human demonstrations is not new. Early works [13, 20, 21] focused on imitation learning using high-level states such as trajectories. Recently, researchers [1, 2, 11, 14, 22, 23, 24, 25, 26, 27, 28, 29, 30, 31, 32, 33] have started exploring raw-pixel inputs with the advancement of deep neural networks. Additionally, the requirement of demonstrations has been reduced by eliminating the need for actions. Recent approaches have explored various one-shot task descriptors, including images [23, 30], language [4, 18, 29, 33], robot video [2, 11, 32], or human video [14, 24]. These methods can be broadly classified into three categories: model-agnostic meta-learning [2, 23, 24, 27, 30], conditional behavior cloning [1, 4, 11, 14, 32], and task graph construction [28, 34]. While significant progress has been made in this direction, these approaches only consider visual observations and primarily focus on simple robotic manipulations such as reach, pick, push, or place. Our dataset offers the opportunity to take a step further by enabling the learning of *hundreds* of skills that require *multi-modal perception* within a single imitation learning model.

**Multi-Modal Learning of Vision and Force**   Force perception plays a crucial role in manipulation tasks, providing valuable and complementary information when visual perception is occluded. The joint modeling of vision and force in robotic manipulation has recently garnered interest within the research community [35, 36, 37, 38, 39, 40, 41]. However, most of these studies overlook the asynchronous nature of different modalities and simply concatenate the signals before or after the neural network. Moreover, the existing research primarily focuses on designing multi-modal learning algorithms for specific tasks, such as grasping [40], insertion [38], twisting [35], or playing Jenga [37]. A recent attempt [42] explores jointly imitating the action and wrench on 6 tasks respectively. Overall, the question of how to effectively handle multi-modal perception at different frequencies for various skills in a coherent manner remains open in robotics. Our dataset presents an opportunity for exploring multi-sensory learning across diverse real-world skills.

| Dataset | # Traj. | # Skills | # Robots | Human Demo | Contact Rich | Depth Sensing | Camera Calib. | Force Sensing |
|---|---|---|---|---|---|---|---|---|
| MIME [17] | 8.30k | 12 | 1 | ✔ | ✗ | ✔ | ✗ | ✗ |
| RoboTurk [16] | 2.10k | 2 | 1 | ✗ | ✗ | ✗ | ✗ | ✗ |
| RoboNet [12] | 162k | N/A | 7 | ✗ | ✗ | ✗ | ✗ | ✗ |
| BridgeData [43] | 7.20k | 4 | 1 | ✗ | ✗ | ✔* | ✗ | ✗ |
| BC-Z [14] | 26.0k | 3 | 1 | ✔ | ✗ | ✗ | ✗ | ✗ |
| RoboSet [18] | 98.5k | 12 | 1 | ✗ | ✔ | ✔ | ✗ | ✗ |
| BridgeData V2 [19] | 60.1k | 13 | 1 | ✗ | ✔ | ✔* | ✗ | ✗ |
| **RH20T** | 110k | 42 | 4 | ✔ | ✔ | ✔ | ✔ | ✔ |

Table 1: Comparison with previous public datasets: "Camera Calib." indicates extrinsic calibration of all cameras and the robot. "✔*" indicates that only a portion of the images are paired with depth sensing. This comparison highlights the comprehensiveness of our dataset, which is the most extensive dataset for robotic manipulation to date.

| Conf. | Robot | Gripper | 6DoF F/T Sensor | Tactile |
|---|---|---|---|---|
| Cfg 1 | Flexiv | Dahuan AG95 | OptoForce | N/A |
| Cfg 2 | Flexiv | Dahuan AG95 | ATI Axia80-M20 | N/A |
| Cfg 3 | UR5 | WSG50 | ATI Axia80-M20 | N/A |
| Cfg 4 | UR5 | Robotiq-85 | ATI Axia80-M20 | N/A |
| Cfg 5 | Franka | Franka | Franka | N/A |
| Cfg 6 | Kuka | Robotiq-85 | ATI Axia80-M20 | N/A |
| Cfg 7 | Kuka | Robotiq-85 | ATI Axia80-M20 | uSkin |

Table 2: Hardware specification of different configurations.

| Conf. | Modal | Size | Frequency |
|---|---|---|---|
| Cfg 1-7 | RGB image | 1280×720×3 | 10 Hz |
| | Depth image | 1280×720 | 10 Hz |
| | Binocular IR image | 1280×720 | 10 Hz |
| | Robot joint angle | 6 / 7 | 10 Hz |
| | Robot joint torque | 6 / 7 | 10 Hz |
| | Gripper Cartesion pose | 6 / 7 | 100 Hz |
| | Gripper width | 1 | 10 Hz |
| | 6DoF F/T | 6 | 100 Hz |
| | Audio | N/A | 30 Hz |
| Cfg 7 | Tactile | 2×16×3 | 200 Hz |

Table 3: Data information of different configurations. The first 9 data modality are the same for all robot configurations. The last data modality of fingertip tactile sensing is only available in Cfg 7.

## 3 RH20T Dataset

We introduce our robotic manipulation dataset, Robot-Human demonstration in 20TB (RH20T), to the community. Fig. 1 shows an overview of our dataset.

### 3.1 Properties of RH20T

RH20T is designed with the objective of enabling general robotic manipulation, which means that the robot can perform various skills based on a task description, typically a human demonstration video, while minimizing the notion of rigid tasks. The following properties are emphasized to fulfill this objective, and Tab. 1 provides a comparison between our dataset and previous representative publicly available datasets.

**Diversity**  The diversity of RH20T encompasses multiple aspects. To ensure task diversity, we selected 48 tasks from RLBench [6], 29 tasks from MetaWorld [7], and introduced 70 self-proposed tasks that are frequently encountered and achievable by robots. In total, it contains 147 tasks, consisting of 42 skills (*i.e.,* verbs). Hundreds of objects were collected to accomplish these tasks. To ensure applicability across different robot configurations, we used 4 popular robot arms, 4 different robotic grippers, and 3 types of force-torque sensors, resulting in 7 robot configurations. Details about the robot configurations are provided in Tab. 2.

To enhance environment diversity, we frequently replaced over 50 table covers with different textures and materials, and introduced irrelevant objects to create distractions. Manipulations were performed by tens of volunteers, ensuring diverse trajectories. To increase state diversity, for each skill, volunteers were asked to change the environmental conditions and repeat the manipulation 10 times, including variations in object instances, locations, and more. Additionally, we conducted robotic manipulation experiments involving human interference, both in adversarial and cooperative settings.

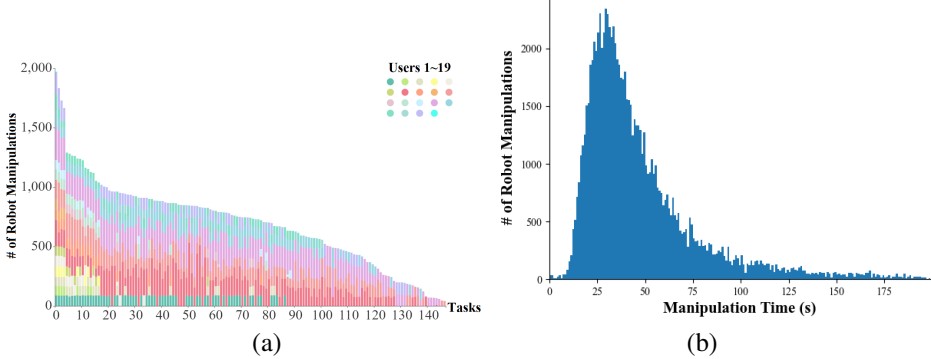

Figure 2: (a) Statistics on the amount of robotic manipulation for different tasks. (b) Statistics on the execution time of different robotic manipulations in our dataset.

**Multi-Modal** We believe that the future of robotic manipulation lies in multi-modal approaches, particularly in open environments, where data from different sensors will become increasingly accessible with advancements in technology. In the current version of RH20T, we provide visual, tactile, audio, and proprioception information. Visual perception includes RGB, depth, and binocular IR images from three types of cameras. Tactile perception includes 6 DoF force-torque measurements at the robot's wrist, and some sequences also include fingertip tactile information. Audio data includes recordings from both in-hand and global sources. Proprioception encompasses joint angles/torques, end-effector Cartesian pose and gripper states. All information is collected at the highest frequency supported by our workstation and saved with corresponding timestamps, and the details are given in Tab. 3.

**Scale** Our dataset consists of over 110,000 robot sequences and an equal number of human sequences, with more than 50 million images collected in total. On average, each skill contains approximately 750 robot manipulations. Fig. 2 (a) provides a detailed breakdown of the number of manipulations across different tasks in the dataset, showing a relatively uniform distribution. Fig. 2 (b) presents statistics on the manipulation time for each sequence in our dataset. Most sequences have durations ranging from 10 to 100 seconds. With its substantial volume of data, our dataset stands as the largest in our community at present.

**Data Hierarchy** Humans can accurately understand the semantics of a task based on visual observations, regardless of the viewpoint, background, manipulation subject, or object. We aim to provide a dataset that offers dense <human demonstration, robot manipulation> pairs, enabling models to learn this property. To achieve this, we organize the dataset in a tree hierarchy based on intra-task similarity. Fig. 3 illustrates an example tree structure and the criteria at different levels. Leaf nodes with a more recent common ancestor are more closely related. For each task, millions of <human demonstration, robot manipulation> pairs can be constructed by pairing leaf nodes with a common ancestor at different levels.

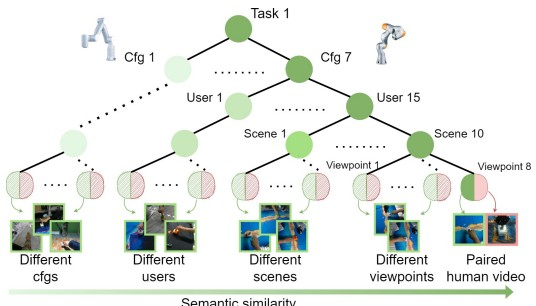

Figure 3: Example of data hierarchy: The leaf nodes in the hierarchy consist of human demonstrations (highlighted in green) and robot manipulations (highlighted in red, only the right-est example is shown in the figure). We can pair a robot manipulation sequence with human demonstration videos captured from different viewpoints, scenes, human subjects, and environments. Zoom in to explore the details of various human demonstrations.

**Compositionality**   RH20T includes not only short sequences that perform single manipulations but also long manipulation sequences that combine multiple short tasks. For example, a sequence of actions such as grabbing the plug, plugging it into the socket, turning on the socket switch, and turning on the lamp can be considered as a single task, with each step also being a task. This task composition allows us to investigate whether mastering short sequences improves the acquisition of long sequence tasks.

## 3.2   Data Collection and Processing

Unlike previous methods that simplify the tele-operation interface using 3D mice, VR remotes, or mobile phones, we place emphasis on the importance of intuitive and accurate tele-operation in collecting contact-rich robot manipulation data. Without proper tele-operation, the robot could easily collide with the environment and generate significant forces, triggering emergency stops. Consequently, previous works either avoid contact [14] or operate at reduced speeds to mitigate these risks.

**Collection**   Fig. 4 shows an example of our data collection platform. Each platform contains a robot arm with force-torque sensor, gripper and 1-2 inhand cameras, 8-10 global cameras, 2 microphones, a haptic device, a pedal and a data collection workstation. All the cameras are extrinsically calibrated before conducting the manipulation. The human demonstration video is collected on the same platform by human with an extra ego-centric camera. Tens of volunteers conducted the robotic manipulation according to our task lists and text description. We make our tele-operation pretty intuitive and the average training time is less than 1 hour. The volunteers are also required to specify ending time of the task and give a rating from 0 to 9 after finishing each manipulation.

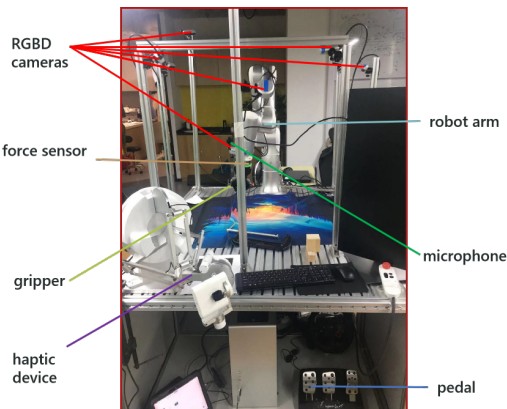

Figure 4: Illustration of our data collection platform.

0 denotes the robot enters the emergency state (e.g., hard collision), 1 denotes the task fails and 2-9 denotes their evaluation of the manipulation quality. The success and failure cases have a ratio of around 10:1 in our dataset.

**Processing**   We preprocess the dataset to provide a coherent data interface. The coordinate frame of all robots and force-torque sensors are aligned. Different force-torque sensors are tared carefully. The end-effector Cartesian pose and the force-torque data are transformed into the coordination system of each camera. Manual validation is performed for each scene to ensure the camera calibration quality. Fig. 5 shows an illustration of rendering different component of the data in a unified coordinate frame and demonstrates the high-quality of our dataset. The detailed data format and data access APIs are provided on our website.

# 4   Discussion and Conclusion

In this paper we present the RH20T dataset for diverse robotic skill learning. We believe it can facilitate many areas in robotics, especially for robotic manipulation in novel environments. The current limitations of this paper are that (i) the cost of data collection is expensive and (ii) the potential of robotic foundation models is not evaluated on our dataset. We have tried to duplicate the results of some recent robotic foundation models but haven't succeeded yet due the limit of computing resources. Thus, we decide to open source the dataset at this stage and hope to promote the

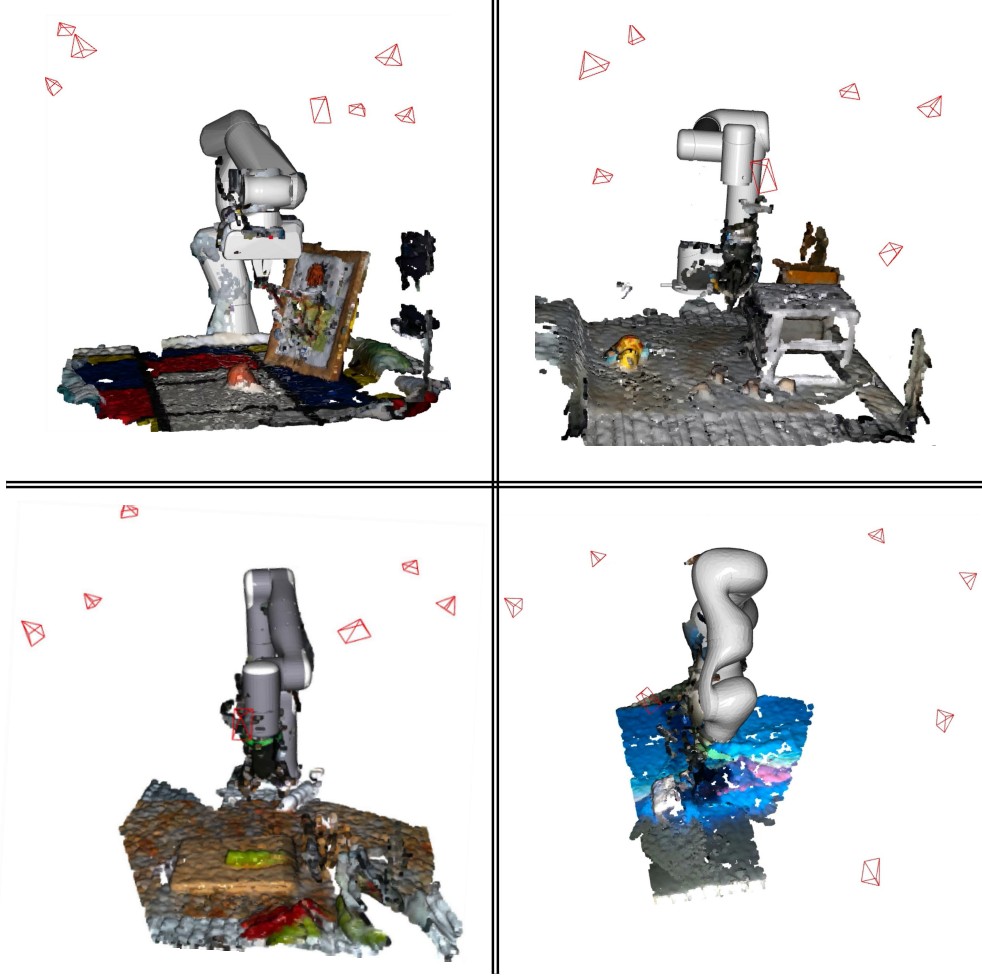

Figure 5: We display the point cloud generated by fusing the RGBD data from the multi-view cameras mounted in our data collection platform. The red pyramids indicate the camera poses. Additionally, the robot model is rendered in the scene based on the joint angles recorded in our dataset. It is evident that all the cameras are calibrated with respect to the robot's base frame, and all the recorded data are synchronized in the temporal domain.

development of this area together with our community. In the future, we hope to extend our dataset to broader robotic manipulation, including dual-arm and multi-finger dexterous manipulation.

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
