# OpenReview forum: "RH20T: A Comprehensive Robotic Dataset for Learning Diverse Skills in One-Shot"
_robot-learning.org/CoRL/2023/Workshop/TGR — CoRL 2023 Workshop TGR Poster_

### Official Review · Reviewer_RNAw · 2023-10-19

**Rating:** 8
**Confidence:** 3

**Review:**

This work collected a dataset comprising over 110,000 contact-rich robot manipulation sequences across diverse skills, contexts, robots, and camera viewpoints, all collected in the real world. Each sequence in the dataset includes visual, force, audio, and action information. The dataset with such diversity and scale can be promising and crucial for the future emergence of foundation models in general skill learning, which shows promising route toward generalist robot.

---

### Official Review · Reviewer_tQXj · 2023-10-20

**Rating:** 9
**Confidence:** 3

**Review:**

This work has assembled a comprehensive multi-view, multi-modal real-world dataset that establishes a strong connection between robot trajectories, human demonstrations, and natural language descriptions. We look forward to the formulation of benchmark tasks based on this dataset.

---

### Decision · Program_Chairs · 2023-10-21

**Decision:**

Accept (Poster)

**Comment:**

Great paper and amazing effort for scaling up real-world robot data!